# Promising M_2_CO_2_/MoX_2_ (M = Hf, Zr; X = S, Se, Te) Heterostructures for Multifunctional Solar Energy Applications

**DOI:** 10.3390/molecules28083525

**Published:** 2023-04-17

**Authors:** Jiansen Wen, Qi Cai, Rui Xiong, Zhou Cui, Yinggan Zhang, Zhihan He, Junchao Liu, Maohua Lin, Cuilian Wen, Bo Wu, Baisheng Sa

**Affiliations:** 1Multiscale Computational Materials Facility, and Key Laboratory of Eco-Materials Advanced Technology, College of Materials Science and Engineering, Fuzhou University, Fuzhou 350100, China; 2College of Materials, Xiamen University, Xiamen 361005, China; 3Department of Ocean and Mechanical Engineering, Florida Atlantic University, Boca Raton, FL 33431, USA

**Keywords:** transition metal dichalcogenides, MXenes, van der Waals heterostructures, first-principle calculations, sustainable energy applications

## Abstract

Two-dimensional van der Waals (vdW) heterostructures are potential candidates for clean energy conversion materials to address the global energy crisis and environmental issues. In this work, we have comprehensively studied the geometrical, electronic, and optical properties of M_2_CO_2_/MoX_2_ (M = Hf, Zr; X = S, Se, Te) vdW heterostructures, as well as their applications in the fields of photocatalytic and photovoltaic using density functional theory calculations. The lattice dynamic and thermal stabilities of designed M_2_CO_2_/MoX_2_ heterostructures are confirmed. Interestingly, all the M_2_CO_2_/MoX_2_ heterostructures exhibit intrinsic type-II band structure features, which effectively inhibit the electron-hole pair recombination and enhance the photocatalytic performance. Furthermore, the internal built-in electric field and high anisotropic carrier mobility can separate the photo-generated carriers efficiently. It is noted that M_2_CO_2_/MoX_2_ heterostructures exhibit suitable band gaps in comparison to the M_2_CO_2_ and MoX_2_ monolayers, which enhance the optical-harvesting abilities in the visible and ultraviolet light zones. Zr_2_CO_2_/MoSe_2_ and Hf_2_CO_2_/MoSe_2_ heterostructures possess suitable band edge positions to provide the competent driving force for water splitting as photocatalysts. In addition, Hf_2_CO_2_/MoS_2_ and Zr_2_CO_2_/MoS_2_ heterostructures deliver a power conversion efficiency of 19.75% and 17.13% for solar cell applications, respectively. These results pave the way for exploring efficient MXenes/TMDCs vdW heterostructures as photocatalytic and photovoltaic materials.

## 1. Introduction

The production of sustainable and renewable energy can effectively tackle both the increasing global energy demand and environmental pollution issues [1]. Harnessing solar energy via photocatalytic hydrogen productions and photovoltaic solar cells have been proposed as promising solutions, which optimize the utilization of solar energy in a cost-effective and efficient way [2,3,4]. The search for efficient photocatalytic and photovoltaic materials is one of the most daunting tasks in the use of solar energy nowadays. Since graphene has been successfully produced and applied [5,6,7,8], great attention has been paid to 2D materials, for instance, transition metal carbides/nitrides (MXenes) [9,10,11], carbonitrides [12] transition metal dichalcogenides (TMDCs) [13,14,15], black phosphorene [16] and silicene [17]. Herein, MXenes, first successfully divested from the MAX phase in 2011 [18,19], can be expressed by M*_n_*_+1_X*_n_*T*_x_* (*n* = 1~3), in which M is early transition metals, X refers to carbon or nitrogen, and T indicates surface terminations, such as hydroxyl groups, oxygen, or fluorine [10,20]. Usually, the functionalization of the MXenes surface makes the modulation of electronic properties more efficient [21,22], which enhances the possibility of solar energy-related applications [23,24,25,26].

On the other hand, the structural formula of TMDCs is generally expressed as MX_2_, where M refers to the transition metal elements (W, Mo, Re, etc.), and X represents the elements (S, Se, and Te). TMDCs materials exhibit a similar three-layer structure, where a transition metal atom single layer is sandwiched between two hexagonal chalcogenide atom planes. TMDCs have garnered significant interest owing to their emergent properties in the realm of light-emitting and photonic devices [27,28,29]. Among the TMDCs family, MoS_2_, MoSe_2,_ and MoTe_2_ stand out with their distinguished physical and chemical properties, such as good stability, flexibility, electronic conductivity, optical, and catalytic properties [30,31,32,33]. However, the applications of transition metal dichalcogenides (TMDCs) in photocatalytic and photovoltaic applications are hindered by limitations such as inadequate spatial separation of electron-hole pairs, substantial photo-corrosion, and high light transmittance [34]. Consequently, a promising approach to enhance electron-hole separation in TMDCs is the construction of heterostructures with 2D semiconducting materials.

To further enhance the properties of 2D materials, vdW heterostructures with weak vdW interactions between vertical layers have been proposed [35,36]. Based on the interlayer coupling effect, 2D vertical vdW heterostructures exhibit distinct advantages derived from each constituent material, thereby yielding unique and promising features [37,38]. MXenes and TMDCs are potential candidates to form vdW heterostructures. For instance, MoSe_2_/Ti_3_C_2_ [39] and MoSe_2_/Ti_3_C_2_O_2_ [40] integrate the properties of their individual components, which show potential applications in photocatalysis, photovoltaics, and optoelectronics. The abundance and possibilities of combining MXenes and TMDCs together promote the investigation of MXene/TMDC vdW heterostructures of general interest and great importance. It is worth noting that the oxygen-functionalized MXenes materials Hf_2_CO_2_ and Zr_2_CO_2_ possess suitable band gaps for solar energy harvesting applications. The construction of heterostructures using M_2_CO_2_ (M = Hf, Zr) and TMDCs MoX_2_ (X = S, Se, Te) with the same hexagonal 2D lattice and restricted lattice mismatch not only serves as a remedy for the aforementioned drawbacks of MoX_2_ materials but also exhibits significant potential for optoelectronic and photocatalytic applications.

In this work, we constructed the M_2_CO_2_/MoX_2_ (M = Hf, Zr; X = S, Se, Te) heterostructures to investigate their photocatalytic hydrolysis and photoelectric conversion mechanism to evaluate the potential in photocatalytic and photovoltaic applications. Using first-principles calculations, we performed a comprehensive study on the structural stability, electronic structure, photocatalytic mechanism, and optoelectronic properties of M_2_CO_2_/MoX_2_ heterostructures. Interestingly, the Zr_2_CO_2_/MoSe_2_ and Hf_2_CO_2_/MoSe_2_ heterostructures have the potential for promising candidates for overall water splitting, owing to their band gaps and band edge positions that are suitable for photocatalytic water splitting. Moreover, the estimated maximum power conversion efficiencies of Hf_2_CO_2_/MoS_2_ and Zr_2_CO_2_/MoS_2_ heterostructures are pretty excellent and are considerably competitive with other existing heterostructures. These findings will diversify catalyst options of MXenes/TMDCs vdW heterostructures for photocatalytic hydrogen production and solar cell energy storage.

## 2. Experimental Section

### 2.1. Computational Details

Our density functional theory (DFT) calculations were performed by using the VASP package [41] together with the ALKEMIE platform [42,43]. The exchange-correlation interactions between electrons were carried out using the projector-augmented wave (PAW) [44] generalized gradient approximation (GGA) of the Perdew-Burke-Ernzerhof (PBE) functional [45]. The cutoff energy for the plane-wave-basis was set to 500 eV. For both geometry optimization and electronic structure calculation, the 2D Brillouin zone was installed with a Γ-centered k point of 12 × 12 × 1 mesh. To eliminate the interactions of periodic two-layer adjacent heterostructures, a vacuum layer larger than 20 Å was added along the z-direction of the 2D models. To have the assurance that the heterogeneous structures were fully optimized, the precision energy and precision force were required to be within 10^−5^ eV and 0.01 eV/Å, respectively. The Grimme’s DFT-D3 method was used to include the long-range vdW interactions [46,47]. The HSE06 hybrid density was functional and was utilized for the precise determination of bandgap values [48]. To verify the lattice dynamic stability of heterostructures, the phono dispersion curves were calculated using Phonopy code [49] with a 3 × 3 × 1 supercell. In order to investigate the thermodynamic stability of M_2_CO_2_/MoX_2_ heterostructures at room temperature (300 K), ab initio molecular dynamics (AIMD) simulations were performed with a supercell of size 3 × 3 × 1 [50,51].

### 2.2. Data Analysis

To quantify the lattice constant differences between different 2D structures, the degree of lattice mismatch K is defined in the theoretical calculation of heterostructures as [52]:(1)$$K=\frac{2\left|a_{1}-a_{2}\right|}{a_{1}+a_{2}}$$
where $a_{1}$ and $a_{2}$ are the lattice constants of the two monolayers forming the heterostructures. To examine the thermodynamic stabilities of 2D heterostructures, the formation energy $E_{f}$ is obtained according to [53]:(2)$$E_{f}=E_{M_{2}{\mathrm{CO}}_{2}{/\mathrm{MoX}}_{2}}-E_{M_{2}{\mathrm{CO}}_{2}}-E_{{\mathrm{MoX}}_{2}}$$
where $E_{M_{2}{\mathrm{CO}}_{2}{/\mathrm{MoX}}_{2}}$ is the total energy of the M_2_CO_2_/MoX_2_ heterostructures, $E_{M_{2}{\mathrm{CO}}_{2}}$ and $E_{{\mathrm{MoX}}_{2}}$ are the total energies of pristine M_2_CO_2_ and MoX_2_ monolayers, respectively. On the other hand, we have also calculated the 2D heterostructure binding energy $E_{b}$ to assess the strength of vdW interactions according to:(3)$$E_{b}=\left(E_{M_{2}{\mathrm{CO}}_{2}{/\mathrm{MoX}}_{2}}-E_{M_{2}{\mathrm{CO}}_{2}{/\mathrm{MoX}}_{2}}^{h}\right)/S_{h}$$
where $E_{M_{2}{\mathrm{CO}}_{2}{/\mathrm{MoX}}_{2}}^{h}$ is the sum of the total energies of M_2_CO_2_ and MoX_2_ monolayers fixed in the corresponding heterostructure lattice, and $S_{h}$ represents the 2D unit cell area. The absorption coefficients of 2D materials α(ω) are derived from [54]:(4)$$\alpha \left(\omega \right)=\sqrt{2}\omega {\left[\sqrt{\varepsilon_{1}^{2}\left(\omega \right)+\varepsilon_{2}^{2}\left(\omega \right)}-\varepsilon_{1}\left(\omega \right)\right]}^{1/2}\varepsilon_{2}$$
where $\varepsilon_{1}$ and $\varepsilon_{2}$ are the real and imaginary parts of the optical dielectric functions, respectively. The carrier mobility of 2D materials μ was calculated by [55]:(5)$$\mu =\frac{2e\hbar^{3}C_{2D}}{3K_{B}T{\left|m^{*}\right|}^{2}E_{1}^{2}}E_{1}$$
where e, ℏ, $C_{2D}$, $K_{B}$, T, $m^{*}$ and $E_{1}$ are the electron charge, reduced Planck constant, 2D elastic modulus, Boltzmann constant, temperature, carrier effective mass, and deformation potential constant, respectively.

## 3. Results and Discussion

First, we analyzed the geometries and electronic structures of MoX_2_ (X = S, Se, Te) and M_2_CO_2_ (M = Zr, Hf) monolayers. The optimized lattice constants using DFT-D3 functionals of the MoS_2_, MoSe_2_, MoTe_2_, Zr_2_CO_2,_ and Hf_2_CO_2_ monolayers are 3.16, 3.29, 3.52, 3.30, and 3.25 Å, respectively. The corresponding projected band structures are illustrated in Appendix A. MoX_2_ monolayers are direct band gap semiconductors, where the conduction band minimum (CBM) and the valence band maximum (VBM) are located at the K point. The HSE06 band gaps of MoS_2_, MoSe_2,_ and MoTe_2_ monolayers are 2.23, 2.00, and 1.63 eV. While Zr_2_CO_2_ and Hf_2_CO_2_ exhibit indirect bandgap semiconductor features, whose band gap values are 1.68 and 1.69 eV, respectively. The calculated results (listed in Appendix A) are consistent with the previous literature [56,57,58,59,60,61,62,63].

Furthermore, by perpendicularly combining the monolayers, M_2_CO_2_/MoX_2_ (M = Hf, Zr; X = S, Se, Te) heterostructures were built. The lattice mismatches observed between the M_2_CO_2_ and MoX_2_ monolayers fall within the reasonable range from 0.1% to 7.9%, which indicates the feasibility of establishing M_2_CO_2_/MoX_2_ heterostructures. Herein, to investigate the stability of M_2_CO_2_/MoX_2_ heterostructures, six distinct stacking configurations were examined. Taking Zr_2_CO_2_/MoS_2_ as an example, Figure 1 displays the top and side views of the diverse stacking structures examined, and Appendix A presents the respective calculated total energies. Details of the most stable structures after structural optimization under the van der Waals correction algorithm are presented in Appendix A. It is noted that stacking II is the most stable model for most systems. The only exception is that stacking IV is the most stable configuration for Hf_2_CO_2_/MoTe_2_ heterostructure. In the following, the most stable stacking configurations were used for further electronic structure calculations. The formation energies presented in Appendix A demonstrate the energetic favorability of these heterostructures, as evidenced by their negative or minimally positive values [64]. Additionally, all heterostructures are classical van der Waals heterostructures, whose optimized interlayer distances and binding energy are around 3 Å and −20 meV/Å^2^ [65], respectively. Moreover, the negative binding energy corresponds to the exothermic reaction, which further confirms the feasibility from the thermodynamics point of view.

In order to further confirm the dynamic and thermal stabilities of the M_2_CO_2_/MoX_2_ heterostructures, we performed phonon dispersion calculations and AIMD simulations. For one thing, Figure 2 illustrates the phonon dispersion curves for the M_2_CO_2_/MoX_2_ heterostructures. Owing to the existence of eight atoms within each unit cell of the heterostructure, a total of 24 spectral lines are generated in Figure 2, encompassing 21 optical modes and 3 acoustic modes. Furthermore, minimal imaginary frequencies are observed from the phonon dispersion curves, which could be eliminated by depositing the M_2_CO_2_/MoX_2_ heterostructures onto suitable substrates or applying slight strain [66,67]. For another, it can be seen from the energy and structure evolutions in Figure 3 that total energies change in small ranges with temperature and atoms vibrate only slightly around the equilibrium position after a simulation time of 9 ps. Overall, M_2_CO_2_/MoX_2_ vdW heterostructures show good lattice dynamic and thermal dynamic stabilities.

To gain a more comprehensive comprehension of the electronic structures, the projected HSE06 band structures of the M_2_CO_2_/MoX_2_ heterostructures are illustrated in Figure 4. It can be seen that Hf_2_CO_2_/MoTe_2_ and Zr_2_CO_2_/MoTe_2_ show direct band gap structures, where both VBM and CBM locate at the M point. On the other hand, the other heterostructures present indirect band gap features. Appendix A lists the calculated band gaps of the most stable configurations for M_2_CO_2_/MoX_2_ heterostructures. As listed in Appendix A, the band gaps of Hf_2_CO_2_/MoS_2_, Hf_2_CO_2_/MoSe_2_, Hf_2_CO_2_/MoTe_2_, Zr_2_CO_2_/MoS_2,_ Zr_2_CO_2_/MoSe_2_ and Zr_2_CO_2_/MoTe_2_ heterostructures are comparatively lower than those of their corresponding monolayers, with values of 1.35, 1.64, 0.66, 1.11, 1.69, and 1.13 eV, respectively. This observation indicates that the electron can be excited and more accessible with less light energy. In Hf_2_CO_2_/MoS_2_ and Zr_2_CO_2_/MoS_2_ heterostructures, the VBM are primarily influenced by the contribution of the M_2_CO_2_ layers, while the CBM are predominantly determined by the MoX_2_ layers. In contrast, the VBM and CBM are dominated by MoX_2_ and M_2_CO_2_ monolayers in the other heterostructures, respectively. Interestingly, all the M_2_CO_2_/MoX_2_ heterostructures are categorized as type-II heterostructures, which enables the effective separation of photo-generated charge carriers [36].

To further explore the charge transfer between M_2_CO_2_ and MoX_2_ monolayers, the charge density difference Δρ can be calculated from the following [68]:(6)$$\Delta \rho =\rho_{M_{2}{\mathrm{CO}}_{2}{/\mathrm{MoX}}_{2}}-\rho_{M_{2}{\mathrm{CO}}_{2}}-\rho_{{\mathrm{MoX}}_{2}}$$
where the $\rho_{M_{2}{\mathrm{CO}}_{2}{/\mathrm{MoX}}_{2}}$, $\rho_{M_{2}{\mathrm{CO}}_{2}}$, and $\rho_{{\mathrm{MoX}}_{2}}$ are the charge densities of the M_2_CO_2_/MoX_2_ heterostructures, M_2_CO_2_ and MoX_2_ monolayers, respectively. The 3D iso-surface and planar average of the charge difference densities along the z-direction are illustrated in Figure 5. Except for Zr_2_CO_2_/MoS_2_ heterostructure, the charges consume in the MoX_2_ sides and accumulate in the M_2_CO_2_ regions. From the Bader charge analysis listed in Appendix A, we found that 0.0066, 0.0104, 0.0157, 0.0094, and 0.0197 electrons transfer from MoX_2_ to M_2_CO_2_ slabs in the Hf_2_CO_2_/MoS_2_, Hf_2_CO_2_/MoSe_2_, Hf_2_CO_2_/MoTe_2_, Zr_2_CO_2_/MoSe_2,_ and Zr_2_CO_2_/MoTe_2_ heterostructures, respectively. On the contrary, for Zr_2_CO_2_/MoS_2_, there are 0.0158 electrons from Zr_2_CO_2_ to the MoS_2_ side. The charge transfer contributes to the formation of built-in electric fields between the monolayers, which drives the photo-generated electrons and holes in opposite directions and further promotes the separation of electrons from holes [69]. It has been observed that the degree of charge transfer between layers exhibits a strong correlation with photocatalytic and photovoltaic activities. The electron localization functions (ELF) further visualize the detailed chemical bonding features in M_2_CO_2_/MoX_2_ heterostructures. Figure 6 presents the 2D contour plots of ELF in the (110) plane. It shows that the ELF bond points of the vdW bonding are about 0.040 for M_2_CO_2_/MoX_2_ heterostructures, which confirms the vdW interaction between M_2_CO_2_ and MoX_2_ layers.

To understand the chemical driving force for the water-splitting reaction, the band edge alignments of the M_2_CO_2_, MoX_2_ monolayers, and M_2_CO_2_/MoX_2_ heterostructures in conjunction with the work functions were investigated, as shown in Figure 7. The energy levels of the VBM and CBM of the MoS_2_, MoSe_2_, Hf_2_CO_2_, and Zr_2_CO_2_ monolayers satisfy the requirements of the redox potential of water splitting. However, the VBM of the MoTe_2_ monolayer is higher than the oxidation-reduction potential. The result agrees well with the previous report [60]. Herein, the work functions for MoS_2_, MoSe_2_, MoTe_2_, Zr_2_CO_2,_ and Hf_2_CO_2_ monolayers are 5.68, 5.06, 4.60, 5.23, and 5.17 eV, respectively. The disparity in the work function across monolayers contributes to the electron transfer in the vdW heterostructures until the Fermi energy level reaches equilibrium, which is beneficial to form built-in electric fields. Herein, Hf_2_CO_2_/MoSe_2_ and Zr_2_CO_2_/MoSe_2_ heterostructures with suitable band edge alignments for water oxidations and reductions, render them effective photocatalysts for overall water splitting.

To obtain the charge carrier transport characteristics, we calculated the carrier effective masses and mobilities in M_2_CO_2_/MoX_2_ heterostructures along the x and y directions with the orthorhombic lattices. We have rebuilt the hexagonal cell to an orthorhombic cell for the carrier mobility calculations, as shown in Appendix A by taking Zr_2_CO_2_/MoS_2_ as an example. The effective masses ($m^{*}$), deformation potentials ($E_{1}$), 2D elastic modulus ($C_{2D}$), and carrier mobilities (μ) are summarized in Table 1. On the one hand, the predicted hole mobilities of M_2_CO_2_/MoX_2_ heterostructures are much larger than the electron mobilities. On the other hand, the carrier mobilities of M_2_CO_2_/MoX_2_ exhibit the characteristic of high anisotropy. Generally, electron mobilities along the x direction are greater than the y direction, and vice versa, hole mobilities along the y direction are greater than the x direction. Therefore, the electrons tend to go through the x direction on the MoX_2_ side, while the holes demonstrate a predilection for migration along the *y*-axis within the M_2_CO_2_ domain of the heterostructures. Interestingly, the hole mobilities of Hf_2_CO_2_/MoS_2_ and Hf_2_CO_2_/MoTe_2_ heterostructures in the y direction have reached 9140.34 cm^2^ V^−1^s^−1^ and 7592.80 cm^2^ V^−1^s^−1^, respectively, which are greater than silicon (~1400 cm^2^ V^−1^s^−1^) [70]. The strong anisotropic carrier mobility in M_2_CO_2_/MoX_2_ heterostructures can reduce the rate of electron-hole recombination, which is favorable to redox reactions and the photoelectric conversion process.

As the optical property is one of the most important indicators for considering materials in solar energy conversion applications, subsequently, we investigated the optical absorption coefficient of the M_2_CO_2_/MoX_2_ vdW heterostructures. Figure 8 depicts the absorption coefficient curves of the M_2_CO_2_/MoX_2_ heterostructures and the corresponding monolayers using HSE06. In general, M_2_CO_2_/MoX_2_ heterostructures exhibit significant enhancement in the optical absorption coefficient (blue curves in Figure 8), featuring multiple high absorption peaks in both the visible and ultraviolet regions. The absorption coefficient of M_2_CO_2_/MoX_2_ is over two-fold higher than that of M_2_CO_2_ and notably exceeds that of the corresponding MoX_2_ monolayers. It is worth noting that Hf_2_CO_2_/MoS_2_ and Zr_2_CO_2_/MoS_2_ heterostructures exhibit stronger responses to infrared light than monolayers as well. Moreover, compared to M_2_CO_2_ and MoX_2_ monolayers, the absorption peaks of the vdW heterostructures undergo a slight redshift due to the narrowed bandgap. Therefore, the formation of heterostructures enhances optical response, showing promising potential for M_2_CO_2_/MoX_2_ heterostructures in photocatalysis and photovoltaic applications.

According to the previous analysis, Zr_2_CO_2_/MoSe_2_ and Hf_2_CO_2_/MoSe_2_ heterostructures are ideal materials for photocatalytic water splitting. The *E*_VBM_ of Zr_2_CO_2_/MoSe_2_ and Hf_2_CO_2_/MoSe_2_ are more positive than the redox potential of O_2_/H_2_O (1.23 eV corresponds to the potential of −5.67 eV at pH = 0), while the *E*_CBM_ is more negative than the redox potential of H^+^/H_2_O (0 eV corresponds to the potential of −4.44 eV at pH = 0). Meanwhile, they are also type-II semiconductors, which avoid the recombination of photo-generated carriers. Therefore, to further reveal the photocatalytic mechanism, we take the Zr_2_CO_2_/MoSe_2_ and Hf_2_CO_2_/MoSe_2_ heterostructures as examples to further analyze the adsorption and dissociation processes of water molecules on the surface of M_2_CO_2_, as presented in Figure 9a. Under light irradiation, electrons in the Zr_2_CO_2_/MoSe_2_ heterostructure are excited from the valence bands to the conduction bands, while an equal amount of holes remain in the valence bands. Immediately afterward, electrons are transferred from the CBM of MoSe_2_ to Zr_2_CO_2_, and holes are from the VBM of Zr_2_CO_2_ to MoSe_2_, thus forming an internal electric field that effectively accelerates the electron-hole recombination. Combined with the projected band diagram, the hydrogen evolution reaction (HER) occurs on the Zr_2_CO_2_ surface and the oxygen evolution reaction (OER) takes place on the MoSe_2_ surface. For Hf_2_CO_2_/MoSe_2_ heterostructure, the photocatalysis mechanism is the same. Here, we analyzed the HER processes on the Zr_2_CO_2_ and Hf_2_CO_2_ surfaces of the Zr_2_CO_2_/MoSe_2_ and Hf_2_CO_2_/MoSe_2_ heterostructures. Based on the crystal structure features, effective adsorption sites A~C and D~F were considered on the surfaces of Zr_2_CO_2_ and Hf_2_CO_2_, as shown in Appendix A. The adsorption models were subjected to complete relaxation and the adsorption energies of water molecules $E_{\mathrm{ads}}^{H_{2}O/\mathrm{surface}}$ were calculated from [71]:(7)$$E_{\mathrm{ads}}^{H_{2}O/\mathrm{surface}}=E_{\mathrm{tot}}^{H_{2}O/\mathrm{surface}}-E_{\mathrm{tot}}^{H_{2}O}-E_{\mathrm{tot}}^{\mathrm{surface}}$$
where $E_{\mathrm{tot}}^{H_{2}O/\mathrm{surface}}$, $E_{\mathrm{tot}}^{H_{2}O}$ and $E_{\mathrm{tot}}^{\mathrm{surface}}$ are the total energy of the absorption system, individual water molecules, and pristine surface, respectively. Herein, the location with the lowest adsorption energy is where the HER takes place. The optimized models and the corresponding adsorption energies after water adsorption at different sites are exhibited in Appendix A. Different absorbed models obtain adsorption energies from −0.061 to −0.338 eV. All the cases with negative adsorption energies indicate that the water molecule adsorption processes are thermodynamically favorable. It is observed that the adsorption energy is lower in the case of oxygen atoms as adsorption atoms, which indicates that water molecules are more stable on the surface of the heterojunction with oxygen atoms. For the Zr_2_CO_2_ and Hf_2_CO_2_ surfaces, the optimal adsorption sites are C and F, where the HER will take place most possibly.

Furthermore, the generation processes of H_2_ on the Zr_2_CO_2_ and Hf_2_CO_2_ surfaces of Zr_2_CO_2_/MoSe_2_ and Hf_2_CO_2_/MoSe_2_ heterostructures have been illustrated in Figure 9b. For Zr_2_CO_2_/MoSe_2_ heterostructure, the isolated H adatoms exhibit an inclination to migrate in closer proximity to one another under a chemical driving force of −0.21 eV. This driving force is termed as the energy differential between two proximal hydrogen atoms (marked as Step-I and set to 0) and two remote hydrogen atoms (marked as Step-II) on the Zr_2_CO_2_ surface. Afterward, H atoms will form H_2_ molecules (marked as Step-III) accompanied by a chemical driving force of −4.48 eV (from −0.21 eV of Step-II to −4.69 eV of Step-III) on the Zr_2_CO_2_ surface, while the system is less energetic and more thermodynamically stable. The generated hydrogen will be departed easily from the surface (marked as Step-IV), as only 0.12 eV energy is required (from −4.69 eV of Step-III to −4.57 eV of Step-IV). On the other hand, the process of producing H_2_ molecules on the Hf_2_CO_2_ surface of the Hf_2_CO_2_/MoSe_2_ heterostructure is similar. The energies taken for the gradual approach of two distant H atoms to form an H_2_ molecule and then to detach from the surface are −0.30, −6.13, and 0.08 eV, respectively.

To evaluate the application of M_2_CO_2_/MoX_2_ heterostructures in solar cells, we conducted an estimation of the power conversion efficiency (PCE, η), which describes the ability of M_2_CO_2_/MoX_2_ heterostructure materials to transform solar energy into electrical energy, proposed by Scharber et al. [72]:(8)$$\eta =\frac{0.65\left(E_{g}^{d}-\Delta E_{c}-0.3\right)\int_{E_{g}}^{\infty }\frac{P\left(\hbar \varpi \right)}{h\varpi }d\left(\hbar \varpi \right)}{\int_{0}^{\infty }P\left(\hbar \varpi \right)d\left(\hbar \varpi \right)}$$
where 0.65 is the band-fill factor, P(ℏϖ) is the AM1.5 solar energy flux at the photon energy ℏϖ, and $E_{g}^{d}$ is the donor band gap. $\Delta E_{c}$ is the donor and acceptor conduction band offset. To better understand the conduction band offset, Appendix A presents the band structures of individual monolayers that have been arranged in the 2D lattice of the corresponding van der Waals heterostructure, based on the band edge data. The donor band gap $E_{g}^{d}$, conduction band offset $\Delta E_{c}$, and calculated PCE (η) of M_2_CO_2_/MoX_2_ heterostructures are listed in Appendix A. Figure 10 shows simulated solar cell PCE as well as the charge carrier transfer route in M_2_CO_2_/MoX_2_ heterostructures. Interestingly, the maximum PCEs of the Hf_2_CO_2_/MoS_2_ and Zr_2_CO_2_/MoS_2_ heterostructures are calculated to be 19.75% and 17.13% (red star highlighted in Figure 10a), respectively. Remarkably, the donor band gaps of Hf_2_CO_2_/MoS_2_ and Zr_2_CO_2_/MoS_2_ heterostructures are in the range of the ideal band gap for the best light absorption characteristics of solar cells [73,74]. As shown in Figure 10b, photon absorptions in M_2_CO_2_/MoX_2_ heterostructures with high PCEs generate excited-free carriers to develop photocurrent more efficiently.

## 4. Conclusions

In conclusion, we have systematically investigated the geometrical structures, electronic structures, optical properties, photocatalytic and photovoltaic applications of M_2_CO_2_ (M = Hf, Zr), MoX_2_ (X = S, Se, Te) monolayers and corresponding M_2_CO_2_/MoX_2_ vdW heterostructures based on density functional theory calculations. Firstly, the most stable configurations of these heterostructures have been determined by the formation energy. The thermal and lattice dynamic stabilities of the M_2_CO_2_/MoX_2_ heterostructures are demonstrated by the negligible fluctuations of total energy and atomic equilibrium positions with temperature during ab initio molecular dynamics simulations, coupled with the scarcity of imaginary frequencies present in the phonon dispersion curves. Secondly, the calculated bandgap values of these heterostructures exhibit a reduced extent in comparison to those of the associated monolayers. It is noted Hf_2_CO_2_/MoTe_2_, and Zr_2_CO_2_/MoTe_2_ heterostructures display direct band gap characteristics, which are favorable for solar light absorption. Moreover, it is found that built-in polarization electric fields generated near the interfaces, as well as the high and anisotropic carrier mobility, can facilitate the photo-generated carrier separation to improve the photoelectric conversion process. In contrast to the M_2_CO_2_ and MoX_2_ monolayers, the M_2_CO_2_/MoX_2_ heterostructures display a heightened optical absorption effect, predominantly within the ultraviolet and visible light spectra. Interestingly, the M_2_CO_2_/MoX_2_ heterostructures all exhibit the intrinsic type-II semiconductors, the VBM and CBM primarily influenced by the contribution of different layers, which effectively hinder the recombination of electron-hole pairs. Lastly, the Zr_2_CO_2_/MoSe_2_ and Hf_2_CO_2_/MoSe_2_ heterostructures are considered highly prospective contenders for water splitting, given their appropriate band gaps and band edge positions that furnish ample driving force for the redox reaction of water. Furthermore, the designed Hf_2_CO_2_/MoS_2_ and Zr_2_CO_2_/MoS_2_ heterostructures can achieve PCE values of 19.75% and 17.13%, respectively. The present study reveals that M_2_CO_2_/MoX_2_ vdW heterostructures are potential candidates for photocatalytic and photovoltaic device applications.

## Figures and Tables

**Figure 1 molecules-28-03525-f001:**
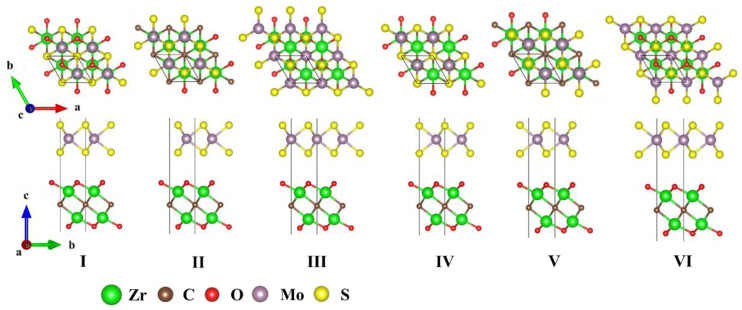
Schematic views of the Zr_2_CO_2_/MoS_2_ heterostructures stacked by varying the position of MoS_2_ layers. The upper layer and the lower layer represent MoS_2_ and Zr_2_CO_2_, respectively. I to VI correspond to the six configurations.

**Figure 2 molecules-28-03525-f002:**
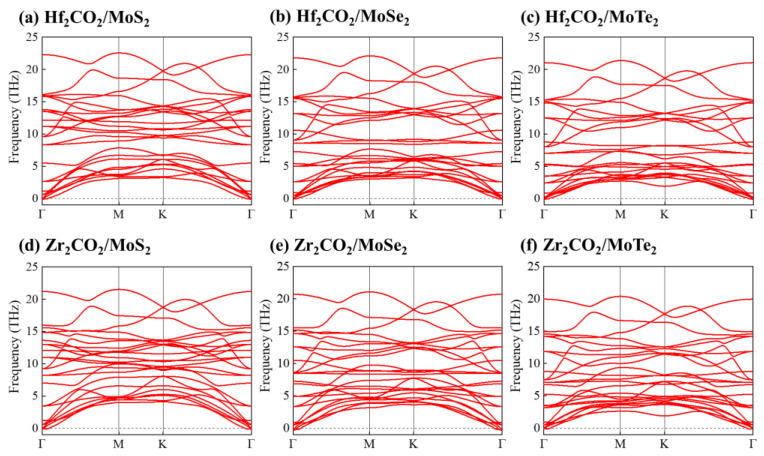
The phonon dispersion curves of (**a**) Hf_2_CO_2_/MoS_2_, (**b**) Hf_2_CO_2_/MoSe_2_, (**c**) Hf_2_CO_2_/MoTe_2_, (**d**) Zr_2_CO_2_/MoS_2_, (**e**) Zr_2_CO_2_/MoSe_2_, and (**f**) Zr_2_CO_2_/MoTe_2_ heterostructures.

**Figure 3 molecules-28-03525-f003:**
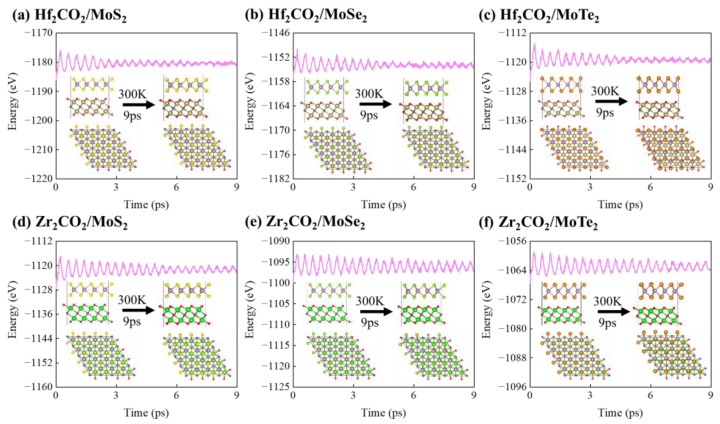
The evolutions of total energy and structure snapshots from AIMD simulations of (**a**) Hf_2_CO_2_/MoS_2_, (**b**) Hf_2_CO_2_/MoSe_2_, (**c**) Hf_2_CO_2_/MoTe_2_, (**d**) Zr_2_CO_2_/MoS_2_, (**e**) Zr_2_CO_2_/MoSe_2_, and (**f**) Zr_2_CO_2_/MoTe_2_ heterostructures.

**Figure 4 molecules-28-03525-f004:**
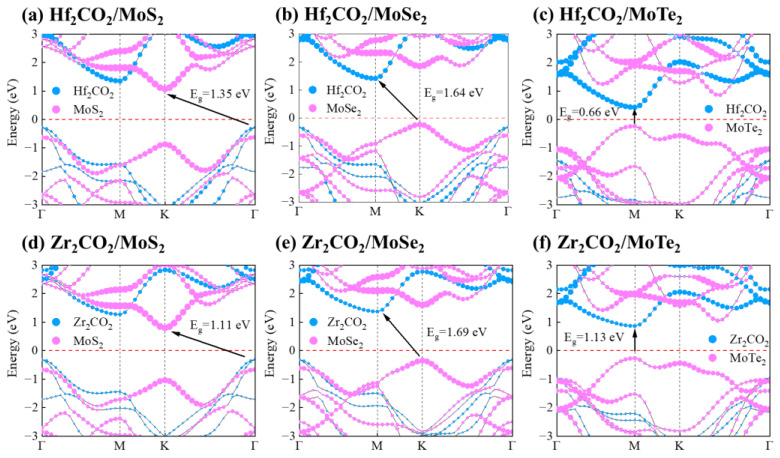
The projected band structures of (**a**) Hf_2_CO_2_/MoS_2_, (**b**) Hf_2_CO_2_/MoSe_2_, (**c**) Hf_2_CO_2_/MoTe_2_, (**d**) Zr_2_CO_2_/MoS_2_, (**e**) Zr_2_CO_2_/MoSe_2_ and (**f**) Zr_2_CO_2_/MoTe_2_ heterostructures using HSE06 functional. The Fermi energy level is set to zero.

**Figure 5 molecules-28-03525-f005:**
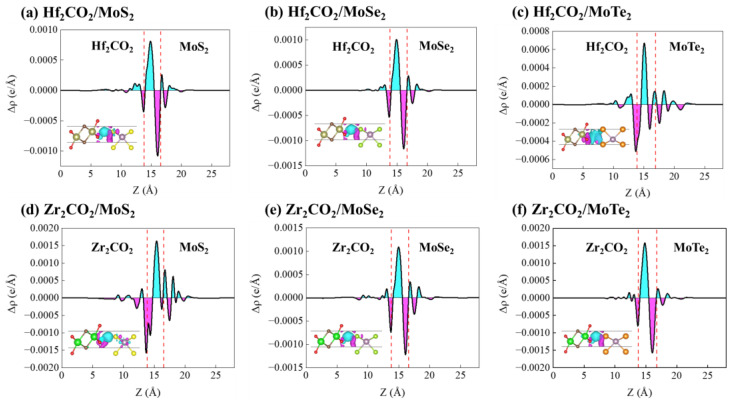
The 3D iso-surfaces and plane-averaged charge density differences along the *z*-direction for: (**a**) Hf_2_CO_2_/MoS_2_, (**b**) Hf_2_CO_2_/MoSe_2_, (**c**) Hf_2_CO_2_/MoTe_2_, (**d**) Zr_2_CO_2_/MoS_2_, (**e**) Zr_2_CO_2_/MoSe_2_, and (**f**) Zr_2_CO_2_/MoTe_2_ heterostructures. The depletion and accumulation of electrons are represented by magenta and blue contours, respectively.

**Figure 6 molecules-28-03525-f006:**
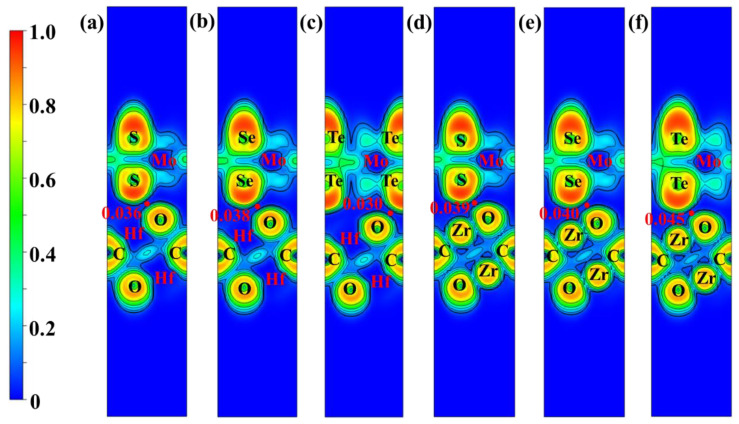
ELF contour plots projected on the (110) plane of (**a**) Hf_2_CO_2_/MoS_2_, (**b**) Hf_2_CO_2_/MoSe_2_, (**c**) Hf_2_CO_2_/MoTe_2_, (**d**) Zr_2_CO_2_/MoS_2_, (**e**) Zr_2_CO_2_/MoSe_2_ and (**f**) Zr_2_CO_2_/MoTe_2_ heterostructures.

**Figure 7 molecules-28-03525-f007:**
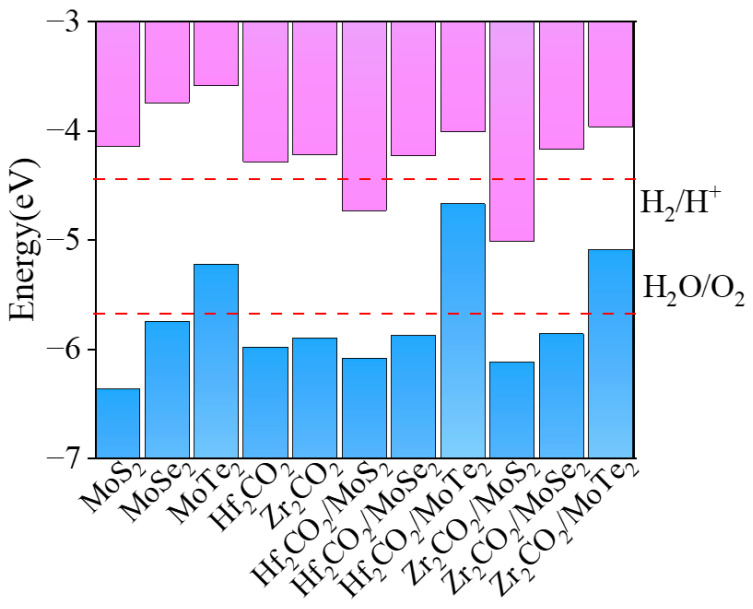
Band edge alignments of the M_2_CO_2_, MoX_2_ monolayers, and M_2_CO_2_/MoX_2_ vdW heterostructures.

**Figure 8 molecules-28-03525-f008:**
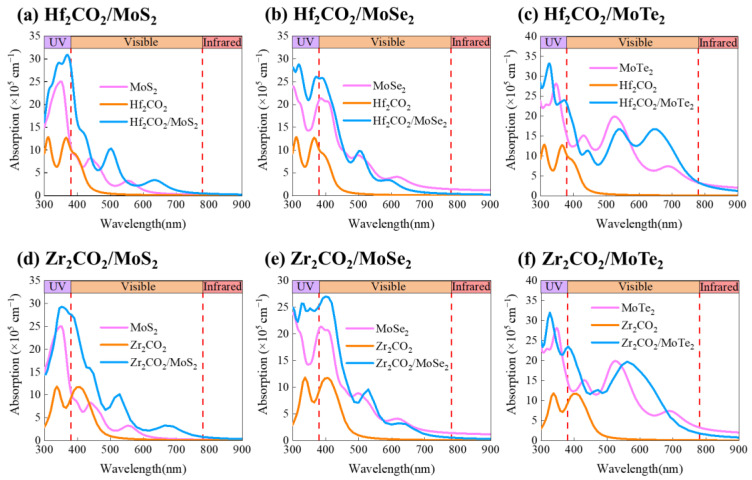
The absorption coefficient of (**a**) Hf_2_CO_2_/MoS_2_, (**b**) Hf_2_CO_2_/MoSe_2_, (**c**) Hf_2_CO_2_/MoTe_2_, (**d**) Zr_2_CO_2_/MoS_2_, (**e**) Zr_2_CO_2_/MoSe_2_, and (**f**) Zr_2_CO_2_/MoTe_2_ heterostructures together with the pristine monolayers using HSE06 functional.

**Figure 9 molecules-28-03525-f009:**
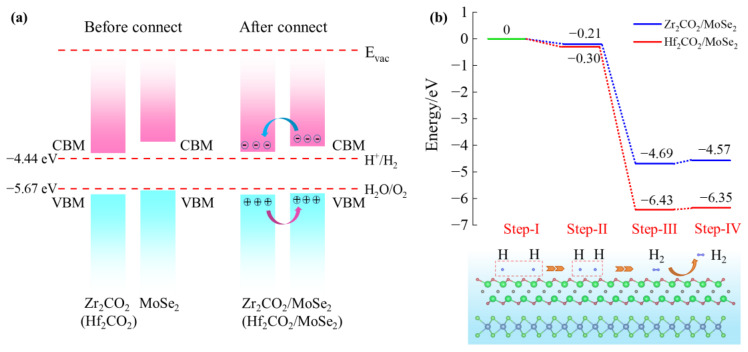
(**a**) The schematic illustrations of the charge transfer paths and overall water splitting of Zr_2_CO_2_/MoSe_2_ (Hf_2_CO_2_/MoSe_2_) heterostructure. (**b**) The generation processes of hydrogen on the Zr_2_CO_2_ (Hf_2_CO_2_) surface of Zr_2_CO_2_/MoSe_2_ (Hf_2_CO_2_/MoSe_2_) heterostructure.

**Figure 10 molecules-28-03525-f010:**
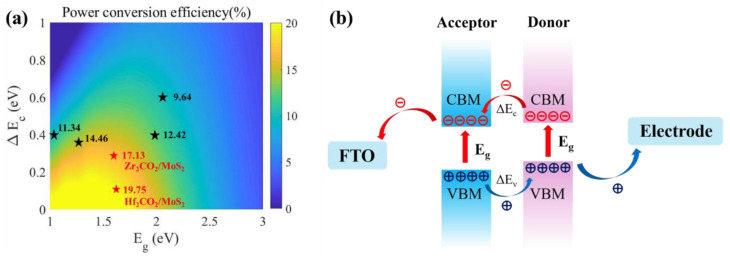
(**a**) A simulated PCE map refers to the donor band gap and conduction band offset of M_2_CO_2_/MoX_2_ heterostructures. (**b**) Schematic illustration of free charge carrier transfer path in M_2_CO_2_/MoX_2_ heterostructure solar cells.

**Table 1 molecules-28-03525-t001:** The effective mass *m** (m_0_), deformation potentials *E*_1_ (eV), elastic moduli *C*_2D_ (N/m), and carrier mobility *μ* (cm^2^ V^−1^s^−1^) of MoX_2_/MoX_2_ heterostructures.

System	Direction	Carrier Type	*E* _1_	*C* _2D_	*m**	*μ*
Hf_2_CO_2_/MoS_2_	x	e	−10.48	428.34	1.16	41.00
		h	2.06	428.34	−0.55	4707.47
	y	e	−9.40	421.71	2.00	16.71
		h	1.33	421.71	−0.61	9140.34
Hf_2_CO_2_/MoSe_2_	x	e	8.44	419.80	0.51	323.12
		h	2.80	419.80	−0.66	1743.19
	y	e	7.63	417.61	2.09	23.01
		h	2.72	417.61	−0.58	2393.98
Hf_2_CO_2_/MoTe_2_	x	e	7.94	409.12	0.56	295.46
		h	−3.75	409.12	−1.30	243.67
	y	e	5.87	404.39	2.22	33.41
		h	−1.78	404.39	−0.49	7592.80
Zr_2_CO_2_/MoS_2_	x	e	−11.47	394.34	0.80	65.10
		h	5.21	394.34	−0.72	396.73
	y	e	−11.49	335.30	1.37	19.08
		h	2.88	335.30	−0.61	1555.99
Zr_2_CO_2_/MoSe_2_	x	e	6.82	368.68	0.76	192.50
		h	1.32	368.68	−0.87	3928.96
	y	e	4.43	392.74	2.68	39.24
		h	2.19	392.74	−0.60	3227.15
Zr_2_CO_2_/MoTe_2_	x	e	10.55	323.53	8.85	0.52
		h	−3.29	323.53	−1.64	155.33
	y	e	6.10	369.17	2.95	16.00
		h	−3.81	369.17	−0.55	1192.02

## Data Availability

The data presented in this study are available on request from the corresponding authors.

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
