# Peer review of "Promising M2CO2/MoX2 (M = Hf, Zr; X = S, Se, Te) Heterostructures for Multifunctional Solar Energy Applications"

_molecules, 2023, doi:10.3390/molecules28083525_

Round 1

Reviewer 1 Report

1. Write the important findings in the conclusion

2. Figure 3 quality is not good. Please correct it.

3. Kindly rewrite/reframe the first two sentences in the introduction part.

4. Kindly incorporate the experimental section after the introduction.

Reviewer 2 Report

1. It is necessary to specify the research objectives.

2. For formulas 1-5 there is no description of physical quantities as well as dimensions.

3. Figure 2 and 4 should be described in more detail.

4. Formulas 7 and 8 need to be explained in the context of the main objectives of the scientific research.

Reviewer 3 Report

In this manuscript, the authors demonstrate the DFT calculation study on the electronic and optical properties of M2CO2/MoX2 vdW heterostructures, as well as their photocatalytic and photovoltalic applications. The authors present a comprehensive study on various properties of M2CO2/MoX2. However, major revisions will be needed for this manuscript to be considered by Molecules.

1. In all atomic schematics throughout this manuscript, there is no M-C bonding shown in the M2CO2 structure and C atoms are shown as isolated layers between M-O layers. Please fix this issue for all figures. Also, the S/Se/Te atom colors are not consistent for all figures. In Figure 1 S/Se/Te are same color, but Figure 3 and Figure 5 they are different color. I would recommend the authors to keep it consistent.

2. The 3D isosurface illustration is confusing. I assume the horizontal direction is the z direction, but no periodicity are shown in the x-y plane, and the atoms in M2CO2 and MoX2 are misaligned. Please revise the figure to make it more accessible.

3. The anisotropic carrier mobilities along x and y directions are interesting. I assume the x and y direction means a and b axes in the 2D plane? If so, I recommned the authors to add clarification to it. Is this anisotropy related to the twist angle between M2CO2 and MoX2? 

4. Figure 11 and the related discuss is the most problematic part in this manuscript. The author constructed a mesoscopic solar cell model with the heterostructure thickness varies from 50 nm to 200 nm. However, all the previous calculation was based on monolayer + monolayer heterostructures. Since the electronic band structures for 2D materials are very sensitive to thickness, the authors need to provide strong justifications on how the 50 nm 2D structure is constructed, and the reason of assuming similar properties for monolayer and bulk. I would simply recommend the authors to remove this part entirely and focus on the photocatalic applications.

Round 2

Reviewer 3 Report

The authors have addressed all comments in the previous review. I recommend to accept the manuscript in present form.